# Heart Failure and Cardiorenal Syndrome: A Narrative Review on Pathophysiology, Diagnostic and Therapeutic Regimens—From a Cardiologist’s View

**DOI:** 10.3390/jcm11237041

**Published:** 2022-11-28

**Authors:** Angelos C. Mitsas, Mohamed Elzawawi, Sophie Mavrogeni, Michael Boekels, Asim Khan, Mahmoud Eldawy, Ioannis Stamatakis, Dimitrios Kouris, Baraa Daboul, Oliver Gunkel, Boris Bigalke, Ludger van Gisteren, Saif Almaghrabi, Michel Noutsias

**Affiliations:** 1Department of Internal Medicine A (Division of Cardiology, Angiology, Nephrology and Intensive Medical Care), University Hospital Ruppin-Brandenburg (UKRB), Brandenburg Medical School Theodor Fontane (MHB), Fehrbelliner Strasse 38, D-16816 Neuruppin, Germany; 2Onassis Cardiac Surgery Center, 50 Esperou Street, Palaeo Faliro, 175-61 Athens, Greece; 3Department of Cardiology, University Heart Center Berlin and Charité-Universitätsmedizin Berlin, Campus Benjamin-Franklin (CBF), Hindenburgdamm 30, D-12203 Berlin, Germany; 4Institute of Medical Psychology, Clinical Psychology and Psychotherapy, Brandenburg Medical School Theodor Fontane (MHB), Fehrbelliner Strasse 38, D-16816 Neuruppin, Germany; 5Faculty of Medicine, Martin-Luther-University Halle-Wittenberg, Magdeburger Strasse 6, D-06112 Halle (Saale), Germany

**Keywords:** cardiorenal syndrome, heart failure, pathophysiology, treatment, prognosis

## Abstract

In cardiorenal syndrome (CRS), heart failure and renal failure are pathophysiologically closely intertwined by the reciprocal relationship between cardiac and renal injury. Type 1 CRS is most common and associated with acute heart failure. A preexistent chronic kidney disease (CKD) is common and contributes to acute kidney injury (AKI) in CRS type 1 patients (acute cardiorenal syndrome). The remaining CRS types are found in patients with chronic heart failure (type 2), acute and chronic kidney diseases (types 3 and 4), and systemic diseases that affect both the heart and the kidney (type 5). Establishing the diagnosis of CRS requires various tools based on the type of CRS, including non-invasive imaging modalities such as TTE, CT, and MRI, adjuvant volume measurement techniques, invasive hemodynamic monitoring, and biomarkers. Albuminuria and Cystatin C (CysC) are biomarkers of glomerular filtration and integrity in CRS and have a prognostic impact. Comprehensive “all-in-one” magnetic resonance imaging (MRI) approaches, including cardiac magnetic resonance imaging (CMR) combined with functional MRI of the kidneys and with brain MRI are proposed for CRS. Hospitalizations due to CRS and mortality are high. Timely diagnosis and initiation of effective adequate therapy, as well as multidisciplinary care, are pertinent for the improvement of quality of life and survival. In addition to the standard pharmacological heart failure medication, including SGLT2 inhibitors (SGLT2i), renal aspects must be strongly considered in the context of CRS, including control of the volume overload (diuretics) with special caution on diuretic resistance. Devices involved in the improvement of myocardial function (e.g., cardiac resynchronization treatment in left bundle branch block, mechanical circulatory support in advanced heart failure) have also shown beneficial effects on renal function.

## 1. Introduction

The network organ interactions of the heart and kidneys are intimately involved in the pathophysiology of cardiorenal syndrome (CRS) [1,2]. In addition, interactions of the heart and kidneys with the central nervous system (CNS) contribute to CRS [3]. The first bibliographic reference to this relationship was made almost 200 years ago, in 1836, by Robert Bright, who observed structural changes in the heart in patients with chronic kidney disease [4]. Although the functional relationship between the heart and the kidneys had long been anticipated, the Working Group of the National Heart, Lung, and Blood Institute tried for the first time to define the syndrome in 2004 [5]. The Working Group stated, “In heart failure, it is the result of interactions between the kidneys and other circulatory compartments that increase circulating volume and symptoms of heart failure and disease progression are exacerbated. At its extreme, cardio-renal dysregulation leads to what is termed “cardio-renal syndrome” in which therapy to relieve congestive symptoms of heart failure is limited by further decline in renal function.” In 2008, the Acute Dialysis Quality Initiative introduced a less cardio-centric definition, dividing the forms of CRS into two categories, cardiorenal and renocardiac syndromes [1]. Currently, five types of the syndrome are classified [6], as summarized in Table 1.

Various organ systems and mechanisms contribute to the pathophysiology of the syndrome, including hemodynamic changes, neurohormonal mechanisms, inflammatory reactions, oxidative stress mechanisms, and various less-defined mechanisms. Various biomarkers assessing heart function and damage, glomerular filtration, and renal tubular damage are currently used or under study to diagnose and assess the prognosis of CRS. Further diagnostic opportunities offered by cardiac ultrasound and magnetic resonance imaging (MRI) can be used to evaluate CRS patients [7]. Various endovascular volume assessment techniques are available today, which can provide important insights into the course of the disease and contribute to clinical decision-making. The pillars of the treatment strategy of the syndrome include decongestion therapy, the use of vasodilators and inotropic agents, and inhibition of RAAS. In addition to clinical signs and symptoms of heart failure, CRS may also contribute to supraventricular arrhythmias, especially atrial fibrillation and sudden cardiac death [8]. CRS has also recently been associated with impaired prognosis in COVID-19 disease [9]. Newer drugs and therapies, such as SGLT2 inhibitors, tolvaptan, and cardiac resynchronization therapy (CRT), have now been suggested as potential therapeutic agents for CRS.

This review aims to summarize pathophysiological concepts, diagnosis, and treatment of CRS, focusing on type 1 and 2 CRS.

## 2. CRS Types and Epidemiological Data

CRS is a group of disorders that bear out the reciprocal relationship between cardiac and renal injury. Different observational and retrospective studies have shown the prevalence and burden of each of the five types of CRS. CRS type 1 is the most common. The nature of epidemiologic data limits clear delineation between cardiorenal syndrome types 2 and 4. Overall, the presence of cardiac or renal dysfunction strongly inhabits a poor prognosis of the contrary organ [10].

Type 1 CRS (acute cardiorenal) is characterized by the acute worsening of cardiac function leading to AKI (Figure 1a) [11]. This type occurs in about 25% of hospitalized cases with acute decompensated heart failure (ADHF) [12]. A preexistent chronic kidney disease (CKD) is common and linked to acute kidney injury (AKI) in 60% of all patients studied. AKI can be considered an independent mortality risk factor in ADHF patients, including those with ST myocardial infarction and/or reduced left ventricular ejection fraction (LVEF) [11].

CRS type 2 is characterized by chronic pathological changes in cardiac function leading to kidney injury or dysfunction, and chronic renal disease has been observed in 45-63% of chronic heart failure (CHF) patients (Figure 1b). However, it may be difficult to classify these patients, which often include those shifting from a clinical condition of type 1 CRS [11]. Of note, the initial decline of glomerular filtration rate (GFR) regularly observed after initiation of heart failure medications (ACEi, ARB, ARNI, SGLT2i) results from the reduction in glomerular pressure, but the decline in GFR either slows or remains unchanged from natural course [13,14].

Type 3 CRS is characterized by acute worsening of kidney function leading to heart disease. A wide spectrum of cardiac dysfunction includes cases with acute decompensated heart failure, acute coronary syndrome, and arrhythmias as defined by the RIFLE (Risk, Injury, Failure, Loss, End-stage kidney disease) and AKIN (Acute Kidney Injury Network) criteria (Figure 1c) [11,15]. AKI actually represents an independent cardiovascular risk factor for mortality in hospitalized patients, especially in those on renal replacement therapy (RRT).

AKI seems to involve almost 70% of patients in ICUs, where 5–25% of patients can develop severe AKI, with mortality rates ranging from 50 to 80% [16]. ADHF still represents the most common acute cardiac dysfunction syndrome worldwide, and it can be defined as new-onset or gradual or rapid worsening of preexistent heart failure with signs and symptoms requiring immediate therapy [17]. Cardiac valvular disease, atrial fibrillation, arterial hypertension, as well as noncardiac comorbidities (renal dysfunction, diabetes, anemia) and medications (especially non-steroidal anti-inflammatory drugs and glitazones) can contribute to ADHF development [17,18]. Renal dysfunction affects mortality rates in ADHF patients from 1.9% (mild renal disease) to 7.6% (severe renal dysfunction) [17].

Type 4 CRS, also defined as chronic renocardiac disease, is characterized by cardiovascular involvement in patients affected by CKD at any stage (Figure 1d). It is well established that renal dysfunction is an independent risk factor for cardiovascular disease with a higher mortality risk for myocardial infarction and sudden death in CKD. A meta-analysis by Tonelli et al. conducted on 1.4 million patients found higher mortality rates for all causes with eGFR decline with relative death odds ratios of 1.9, 2.6, and 4.4 for GFR levels of 80, 60, and 40 mL/min, respectively [19]. The largest epidemiological study was actually performed by Go et al. on over 1 million people; cardiovascular risk was found particularly evident in patients with stages IIIb-IV (according to the K/DOQI CKD classification) renal disease and in those who underwent RRT (hemodialysis, peritoneal dialysis, and transplantation) [20]. The Chronic Renal Insufficiency Cohort (CRIC) Study focused on 190 patients presenting stage III to end-stage renal disease and performing serial echocardiographic exams; in the 2-year evaluation period in which patients shifted from stage V to end-stage renal disease, the EF dropped from 53 to 50%; therefore, they found that the number of subjects with EF <50% increased by 20% [21].

Type 5 CRS takes place when cardiac and renal injury occur at the same time as it occurs in sepsis and in systemic inflammatory response syndrome (SIRS) (Figure 1e) [18]. Type 5 CRS is involved in COVID-19-associated CRS [9]. Type 5 CRS is a recently defined clinical syndrome; thus, solid epidemiological data are not available.

## 3. Pathophysiology

Various mechanisms playing a role in the onset of CRS have been identified, while several others are still under study. There is a close relationship between heart failure endpoints (re-hospitalization due to heart failure, mortality) with the degree of renal impairment [22,23]. The hemodynamic changes associated with acute heart failure may play an important role in the development of CRS. The initial hypothesis, that reduced cardiac output leads to hypoperfusion and ischemia of the kidneys, which in turn leads to renal dysfunction and activation of neurohormonal mechanisms [24], does not necessarily reflect the multifaceted complexity of CRS completely. While an association between AKI and decreased cardiac index in AHF was confirmed in one trial on cardiogenic shock patients [25], the ESCAPE trial did not confirm a correlation between hemodynamic changes measured by right heart catheterization and the development of renal function in acute decompensated heart failure (ADHF) [22]. The substantial diversity of the clinical phenotypes included or excluded in the studies (i.e., acute heart failure with the need for inotropic support) may partly explain these inconsistent data. Furthermore, a majority of creatinine changes are related to hemodynamic effects at the glomerulus, which may be reversible and of largely benign nature of the acute hemodynamic changes. In line with this concept, temporal deteriorations of renal function associated with aggressive diuresis in AHF may be dissimilar from traditional causes of AKI [26,27]. Further evidence from heart failure medication trials confirms the necessity of differentiation between acute hemodynamic effects at the glomerular pressure, which regularly occurs after initiation of ACE-i, ARB, ARNI, and SGLT2i without detrimental prognostic relevance, and genuine kidney injury or progression of kidney disease with tubulointerstitial fibrosis [14,28].

One further pathophysiologic concept is that increased central venous pressure and increased intra-abdominal pressure may be decisively involved in the development of CRS in patients with cardiac impairment. The kidneys require continuous and abundant perfusion. Necessary for their function is sufficient filtration pressure, depending on the large difference between driving arterial pressure and venous outflow. The substantial increase in venous pressure and intra-abdominal pressure in heart failure patients also leads to renal venous congestion and a decrease in filtration pressure [29,30]. Consistent with this hypothesis, the development of AKI was associated with persistently elevated central venous pressure in a further investigation [31]. Right atrial pressure was the only hemodynamic factor associated with baseline renal function in a recent post hoc analysis of the ESCAPE study [22]. The excessive activation of mechanisms, such as the renin–angiotensin aldosterone system (RAAS), the activation of the sympathetic nervous system (SNS), and the secretion of antidiuretic hormone (ADH), contribute to the further deterioration of renal function.

Decreased cardiac output activates neurohormonal pathways to increase tissue perfusion. Excessive activation of the RAAS system, and in particular high levels of angiotensin II, has been shown to have negative effects on both the heart and the kidneys. The vasoconstrictive capacity of angiotensin II significantly increases the afterload of the left ventricle, while at the same time, the continuous vasoconstriction of the coronary arteries can lead to myocardial ischemia and endothelial dysfunction, especially after myocardial infarction [32]. Angiotensin ΙΙ also induces left ventricular hypertrophy and remodeling [33]. It can impair the balance of coagulation and fibrinolysis and, in combination with the activation of the sympathetic nervous system, is a pro-inflammatory factor and activator of oxidative stress [34,35]. Angiotensin II-induced renal injury is hypothesized to be caused by increased systemic pressure or even increased intrarenal vasoconstriction, leading to decreased perfusion of the renal tissue and is associated with ischemic lesions [36,37]. Angiotensin II has also been shown to increase proteinuria, which is blamed for intrarenal damage [38,39]. Its action as a pro-inflammatory agent and activator of oxidative stress and its effect on renal fibroblasts and various other biochemical pathways are also implicated as mechanisms responsible for renal damage from increased angiotensin levels [40,41]. Activation of the SNS has also been linked to the pathophysiology of both CRS and heart failure. Activation of the SNS leads to an increase in catecholamines in the peripheral blood, resulting in an increase in blood pressure and heart rate, which leads to increased cardiac workload and, ultimately, a decrease in cardiac output, peripheral perfusion, and further deterioration of cardiac function [42].

Further mechanisms and molecular pathways, such as inflammation, oxidative stress, and imbalance of regulation mechanisms, such as ADH, have been suggested for the pathophysiology of CRS type 5. For COVID-19 disease, a recent systematic review has elucidated the detrimental prognostic impact of CRS in 637 patients (66.2% males) included in 15 studies with CRS or evidence of both cardiac and renal complications following SARS-CoV-2 infection [9]. Cardiac complications included myocardial injury, heart failure, arrhythmias, or myocarditis and cardiomyopathy, while renal complications manifested with AKI with or without oliguria. Heart failure patients with a diagnosis of CRS or evidence of both cardiac and renal complications had more severe disease and poorer prognoses. Patients with cardiorenal injury were often associated with significantly elevated levels of inflammatory markers (CRP, PCT, IL-6). In addition to these SIRS-related mechanisms, the direct cytotoxic effects of the viral infection in renal and myocardial cells are discussed. The involvement of either CRS or concurrent cardiac and renal complications had a significant impact on the severity of the disease and the mortality rate among patients with COVID-19 infection [9].

## 4. Diagnosis

Establishing the diagnosis of CRS requires various tools, including non-invasive imaging modalities, adjuvant volume measurement techniques, invasive hemodynamic monitoring, and biomarkers.

### 4.1. Traditional Diagnostic Methods

GFR remains the gold standard for assessing renal function. However, measuring true, real-time GFR remains difficult in the setting of ADHF or CHF because formulas estimating GFR have been validated when creatinine is in a steady state. Moreover, creatinine represents an imperfect surrogate that entails important limitations. First, creatinine reflects only GFR and not tubular injury directly, whereas tubular injury may help to better predict and characterize AKI and the progression of chronic kidney damage [43]. In addition, creatinine presents a relatively belated rise on the occasion of an episode of AKI. In fact, serum concentration of this marker begins to rise many hours after AKI stabilization, when very little can be done to avoid or counteract the renal worsening in a timely manner [6]. Furthermore, creatinine levels are also influenced by a series of variables such as age, sex, ethnicity, and muscle mass. Consequently, in patients hospitalized with heart failure, especially women and the elderly, who often have decreased muscle mass, seemingly low Cr levels may cause an under-recognition of renal insufficiency [44]. In addition, under-appreciation of the exponential relationship between creatinine and GFR, where small elevations in creatinine in the near-normal range can indicate large reductions in GFR, may further cause the under-recognition of new-onset renal impairment. A kinetic study has also shown significant variability in and a low overall rate of increase of creatinine after AKI, with substantial increases often not being observed until 48–72 h after the initial damage; also, a new steady state is sometimes not reached for up to 7 days, rendering creatinine a relatively belated marker of AKI [45].

A diagnosis of CRS type 2 should be based on a clinical picture of either CHF with preserved or reduced LVEF on echocardiography, joined with biochemical signs of renal dysfunction, the onset or progression of which is reasonably secondary to congestive heart failure (HF). The findings indicative of renal dysfunction in the context of a CRS type 2 should include increased Cr or, in subjects with poor muscle mass, a Cr value in the near-normal range, provided that it is associated with low values (<60 mL/min/1.73 m²) of eGFR, calculated using the Modified Diet in Renal Diseases study (MDRD) or Cockcroft–Gault equations. Further laboratory findings that are useful for a better diagnostic definition of renal damage are represented by the coexistence of albuminuria or anemia, or both. Cardiac index and total peripheral resistance as assessed by pulmonary artery catheter are hypothetically important measures to evaluate the hemodynamic compromise involved in CRS. However, no prognostic benefit could be confirmed in the ESCAPE trial by pulmonary artery catheter monitoring [22].

### 4.2. Renal Biomarkers

Biomarkers in the context of CRS can be primarily divided into three groups; biomarkers of renal tubular injury, markers of glomerular filtration and integrity, and finally, cardiac biomarkers. Biomarkers may be important for the early identification of renal or cardiac injury and the prognostic evaluation of CRS. Biomarkers may also be helpful in the discrimination of various CRS phenotypes and in guiding targeted therapeutic interventions.

Albuminuria and cystatin C (CysC) are the biomarkers of glomerular filtration and integrity in CRS. Albuminuria also has an adverse prognostic value for cardiovascular death, all-cause death, and heart failure re-admissions [5,6]. The presence of endothelial dysfunction, elevated glomerular pressure, and inflammation results in increased excretion of albumin by causing damage to the glomerular membrane [7]. Albuminuria is associated with an increased risk of mortality, heart failure-related hospitalizations, and clinical, echocardiographic, and circulating biomarkers of congestion [46].

Among the markers of renal function applicable to the CRS studies, CysC is one of the most widely exploited as a diagnostic tool for use as a replacement or supplement for creatinine. This protein, with a chain structure consisting of 120 amino acids, is present in almost all tissues and body fluids. Urine Cystatin C is a marker of proximal tubule injury/dysfunction since it is fully metabolized in the proximal tubule except with renal injury or dysfunction. On the other hand, serum CysC levels reflect glomerular filtration [47,48]. Furthermore, CysC may be influenced by factors such as body composition [49]. CysC is a cysteine proteinase inhibitor that might aid in overcoming some of the limitations related to the use of Cr serum levels for the estimation of GFR. Moreover, these levels are less dependent on body mass, nutritional status, or age [50,51]. However, the relatively high cost of measuring CysC, when compared to Cr, contributes to its still limited use in routine clinical practice.

Numerous other biomarkers have been evaluated for the accurate detection of AKI and/or for the prediction of the progression of CKD. While these novel biomarkers may contribute to our pathophysiological understanding of AKI and cardiorenal disease, establishment in routine clinical practice remains to be determined, partly due to higher costs associated with their measurement and the lack of diagnostic or prognostic relevance or superiority compared with established biomarkers. Recently, a prognostic superiority of novel biomarkers compared with established measures (such as creatinine) was not confirmed for NGAL (neutrophil gelatinase-associated lipocalin) for worsening renal function (WRF) and AHF [26,52,53,54,55,56,57,58,59,60,61,62,63,64], nor for kidney injury molecule-1 (KIM-1) [27].

### 4.3. Cardiac Biomarkers

Finally, cardiac biomarkers are also important in CRS. NT-proBNP is a widely used cardiac biomarker for the diagnosis and prognostic stratification of heart failure [65]. ST2 is another protein that offers additional value to natriuretic peptide levels in predicting HF-related hospitalizations and deaths. Notably, ST2 levels are not affected by renal function [66]. Troponins also have prognostic implications when elevated in AHF, even in the absence of underlying coronary artery disease. Elevated troponin levels are also associated with declining GFR and higher mortality [67].

### 4.4. Imaging of the Heart and the Kidneys

Trans-thoracic echocardiography and renal ultrasound provide important information on heart and kidney failure in patients with cardiorenal syndrome and are the imaging methods of choice in the diagnostic work-up [68]. Ultrasound imaging is widely available, non-invasive, and safe. Echocardiography can identify impairment of LVEF, regional wall motion abnormalities (i.e., hypokinesia, akinetic regions suggestive of ischemic condition), left ventricular hypertrophy, valvular stenosis and/or regurgitation, pericardial effusion, aortic aneurysms or dissection. In the context of echocardiographic work-up of CRS, evaluation of the inspiratory collapse of the inferior vena cava is useful for the evaluation of volume status (i.e., congestion) [11,69].

Kidney ultrasound can exclude obstructive uropathy as a cause of renal failure [7]. Kidney ultrasound in CRS usually shows normal or larger renal dimensions with a preserved cortical–medullary ratio, while color Doppler evaluation reveals regular intraparenchymal blood flow, often associated with a raised resistance index (>0.8 cm/s) [11,69]. However, no reliable distinction between inflammation and fibrosis can be identified by echogenicity [70]. Doppler sonography can quantify renal blood flow and intrarenal hemodynamics, which reflect renal dysfunction and/or microstructural alterations. An increased renal resistive index is associated with impaired prognosis in numerous renal disorders [71,72].

Computer tomography (CT) of the heart may be useful for the evaluation of coronary artery calcification (CAC) indicating coronary artery disease. CAC is an independent predictor of cardiovascular morbidity and mortality in CKD patients [73]. CT techniques using iodinated contrast agents are largely not feasible in CRS patients in light of the substantial harm of additional post-contrast AKI [74]. CT of the kidneys and of the urinary tract is useful in the diagnosis of urinary tract obstruction. However, contrast-based CT studies are of limited use in CRS because of the potential complications on renal function [7].

Cardiac magnetic resonance imaging (CMR) is a gold standard approach for the accurate assessment of biventricular function, ventricular geometry and mass, valvular pathologies, pericardial effusion, and of myocardial structure, including the localization and quantification of myocardial scars and inflammation [75,76]. Additionally, multiparametric MRI algorithms based on pre- and post-contrast T1 relaxation time can calculate the extracellular volume fraction (ECV) for the detection of diffuse myocardial fibrosis [77]. Of note, contrast-enhanced MRI is limited in CRS patients since severely impaired renal function (GFR < 30 mL/min/1.73 m²) is associated with nephrogenic systemic fibrosis when using gadolinium [78]. In CRS, pathogenic and diagnostic insights have risen from magnetic resonance imaging (MRI) studies of the kidneys. In conjunction with the high diagnostic value of MRI, comprehensive protocols enabling MRI of the heart and the kidneys simultaneously have been initiated [68]. Functional MRI of the kidneys has gained interest recently, especially in the detection of early changes in AKI and for the prediction of progression and chronic renal failure [79]. Such comprehensive MRI approaches have also been endorsed for combined heart and brain MRI evaluations in light of the pathogenic interactions between the heart and brain in various cardiovascular diseases [80]. Breidthardt et al. reported a close association between prolonged renal T1 relaxation in HF patients with renal impairment and that renal T1 relaxation was associated with classic cardiovascular risk factors [81]. Prevention programs have shown beneficial effects both on cardiovascular endpoints and on dementia, highlighting the interactions between cardiovascular diseases and cognitive function [82]. Simultaneous MRI assessment of all three organs, heart, kidneys, and brain, has been proposed as an “all-in-one-MRI-examination” mode for the additional detection of brain lesions, also limiting the total amount of gadolinium contrast agent [83]. CRS is characterized by multiple alterations of myocardial function, renal function, fluid balance, and neural pathways. Its effects on the peripheral and central nervous system contribute to dysregulation of the sympathetic nervous system, autonomic reflexes, and fluid balance control [3]. The statistical association of altered mental status in cardiogenic shock with adverse prognosis confirmed that hypoperfusion is a mechanistically relevant issue for the decline in mental function [25]. Non-invasive imaging with a low harm profile may be of paramount importance for deciphering the complex, interrelated pathogenic pathways comprising the heart, kidney, and nervous system in the various CRS subtypes and may be pivotal for future improved clinical management strategies.

Functional magnetic resonance (MR) imaging of the kidneys has gained interest recently, especially for the detection of early changes in AKI or for the prediction of the progression of CKD. The application of these methods to CRS is fairly novel. Blood oxygen level-dependent (BOLD) MR imaging of the kidneys in mice with experimental myocardial infarction revealed that R2* in the kidney increased after induced myocardial infarction and that the response was higher in animals with larger infarcts and over time [84]. These BOLD MR imaging findings were associated with the expression of hypoxia-inducible factor-1alpha (HIF-1alpha), an independent marker of renal hypoxia. In addition, they showed evidence of renal injury by using a kidney injury marker, kidney injury molecule-1 (KIM-1). The results of their study support the use of renal BOLD MR imaging in subjects with heart failure, in whom the risk of subsequent renal ischemia and/or hypoxia is known to exist [84]. These results, along with those of other recent reports [85], suggest that functional imaging methods might play a rising role in evaluating changes in both the primary and secondary organs involved in complex disease processes such as CRS. The availability of such methods could facilitate translation to clinical evaluation and improve understanding of the complex pathophysiology [85].

## 5. Treatment Strategies

Cardiorenal syndrome (CRS) is characterized by complex interactions between different physiological systems, whereby dysfunction of the heart negatively affects kidney function and vice versa. There is emerging evidence on the beneficial effects of heart failure treatment also on CRS outcomes. Current treatment options for both HF and CRS include beta-adrenergic blockers (BB), diuretics/ultrafiltration, angiotensin-converting-enzyme-inhibitors (ACEI), angiotensin-receptor-blockers (ARB), mineraloreceptor antagonists (MRA), angiotensin-renin-neprilysin-inhibitors (ARNI), sodium-glucose co-transporter 2 inhibitors (SGLT2i) and cardiac resynchronization (CRT) (Figure 2). Recently, Patel et al. showed that patients with HFrEF and CKD as comorbidity are not optimally treated with the available armamentarium of heart failure medication, even at levels of eGFR at which kidney dysfunction would not be a contraindication for these heart failure substances [23]. In ADHF patients, the addition of acetazolamide to loop diuretics increases successful decongestion and is associated with better diuretic efficiency, as shown in the recently published ADVOR trial [86]. Additional acetazolamide treatment was well tolerated; however, it had no effects on mortality, worsening kidney function, and hypokalemia [86]. Since the pathogenesis of CRS is not yet fully understood, further research on the pathophysiology of the disease is pertinent. The prognostic impact of CRS among patients with COVID-19 infection highlights the need for special assessment and management of CRS in patients with certain comorbidities [9]. The detailed deciphering of such complex interactions may lead to future developments targeting specific pathways, certain CRS types, and leading comorbidities in terms of a personalized and precision medicine approach [87,88].

### 5.1. Regulation of Volume Status

Volume management and diuretics have a narrow therapeutic path in AHF and CRS. Diuretics play a key role in treating CRS since they quickly alleviate symptoms caused by fluid expansion in the patient. They positively affect hypertension, increased intra-abdominal pressure, and renal congestion [30]. Diuretics and/or ultrafiltration are effective for the treatment of ADHF and CRS [89]. In patients with type 1 and type 2 CRS, improvement in renal function is accompanied by improvement in cardiac function [90]. Maintaining hemodynamic stability and guaranteeing tissue perfusion are the key points to prevent type 5 CRS in the hyperacute phase of sepsis, together with fluid control and correct antibiotic treatment. Fluid therapy must be carefully controlled to avoid fluid overload and other iatrogenic complications [2]. Even though loop diuretics are frequently prescribed to treat CRS, there is insufficient clinical data on whether this treatment option reduces short- or long-term mortality or prevents re-hospitalizations and mortality. Patients treated with diuretics must be monitored closely as the use of diuretics may lead to electrolyte imbalance, negatively affect renal function, lead to an excessive decrease in the circulating volume of fluid, and disrupt neurohormonal balance [91,92]. Since diuretics affect neurohumoral activation as well as renal and systemic hemodynamics, inadequate dosage may cause progressive kidney failure. If patients develop diuretic resistance, impaired renal function, and/or persistent fluid expansion, ultrafiltration may be an additional therapeutic strategy. A practical help in the case of increasing retention values with increased doses of loop diuretics is the option of reducing the diuretic dosage again while strictly limiting the amount of fluid consumed (e.g., 1–1.5 L/day). For the sake of completeness, it should be noted that deterioration in renal function with recompensation is not necessarily associated with a poorer prognosis but can also be an expression of hemoconcentration [93]. A further increase in the dose of already higher-dosed loop diuretics does not lead to further natriuresis (“ceiling effect”) but to more side effects. Because of its superior oral bioavailability and longer half-life, torasemide should be preferred to furosemide and should be dosed at least twice daily [94]. In order to achieve a net natriuretic effect, sequential nephron blockade can be initiated by the additional administration of a thiazide or thiazide-like diuretic. This serves to avoid compensatory sodium reabsorption in the distal tubule. Electrolytes (particularly sodium and potassium) should be closely monitored during sequential nephron blockade. This is especially true when triple diuretic therapy is used, such as with a loop diuretic, thiazide diuretic, MRA, or diuretics in combination with SGLT2i. Carbonic anhydrase inhibitors should be avoided because of the risk of severe metabolic acidosis. Early ultrafiltration seems to improve renal outcomes in septic shock patients, but these data have to be confirmed in further clinical trials [95].

### 5.2. Heart Failure Medication

Heart failure with reduced ejection fraction (HFrEF) is routinely treated with BB, ACEI, ARB or ARNI, MRA, and SGKT2i with class I level of evidence [96]. BB have been widely used for the treatment of chronic HF, with evidence for substantial improvement of HF prognosis [96]. However, BB therapy is not an established treatment option for patients with ADHF and in CKD patients without heart failure [97]. Inhibition of the renin–angiotensin–aldosterone system (RAAS) by ACEI, ARB, or ARNI, and MRA is a mainstay of HF treatment leading to improvement of prognosis [96]. ACE-I do not slow the decline of GFR in HFrEF [98]. The evidence on the effects of the ARNI sacubitril/valsartan, in addition to its proven benefits for heart failure outcomes, including both CHF and AHF with (HFrEF) [99,100], has shown that ARNI preserve renal function and inhibit the progressive decline of GFR associated with HF more effectively than ACEi and ARB [39,98,101,102]. However, we lack solid data on the use of ARNI in patients with eGFR <30 mL/min/1.73 m^2^ [103].

### 5.3. Novel Treatment Concepts

Tolvaptan, an oral vasopressin V2-receptor antagonist, was developed for the improvement of ADHF outcomes by targeting the reduction of volume overload and avoiding exuberant intravascular volume depletion, thereby inhibiting harm to renal function. However, the EVEREST trial did not confirm a reduction in long-term mortality or heart failure-related morbidity in ADHF patients [104]. RAAS inhibition may also be associated with an increased risk of hyperkalemia, particularly in patients with an eGFR < 30 mL/min/1.73 m^2^. For this reason, about 80% of patients with renal failure corresponding to CKD stages 3 and 4 (eGRF < 60 mL/min/1.73 m^2^) discontinue treatment with RAAS inhibitors or only receive submaximal doses [105]. However, discontinuation of ACE-I or ARB therapy is associated with a higher risk of mortality and with major cardiovascular events (MACE major adverse cardiac events) [106]. Finerenone, a new, non-steroidal, selective MRA with higher receptor specificity than spironolactone and eplerenone, is now available. In two randomized, double-blind, placebo-controlled, multicenter phase III studies, a benefit regarding cardiovascular and renal events was proven in patients with diabetic nephropathy and albuminuria [107,108,109]. Further studies with finerenone are currently underway, examining the effects in CKD patients without diabetes and in patients with heart failure. If hyperkalemia occurs despite a low MRA dosage (e.g., 25 mg spironolactone or eplerenone; 10 mg finerenone) and renal acidosis can be ruled out, potassium-lowering drugs can be administered if necessary in order to be able to carry out therapy with MRA [110].

A recently established pilar of pharmacological HF therapy are the SGLT2i, initially introduced as oral antidiabetic agents, both in HFrEF and HFpEF [96,111,112,113,114,115]. The period of time between initiation and significant clinical benefit of SGLT2i is low (28 days), as shown in the DAPA-HF trial [116]. There is no evidence for rising adverse effects as AKI with SGLT2i [117]. Recently, empagliflozin treatment has also shown beneficial effects in ADHF patients, increasing diuretic efficiency without affecting markers of renal function or renal injury with an overall good safety profile [118,119]. Beneficial effects of dapagliflozin have been confirmed in heart failure patients with mild and preserved ejection, mainly driven by a reduction in worsening heart failure and re-hospitalization events in the DELIVER trial [114]. These data have substantiated evidence of SGLT2i as the fourth pillar of pharmacological heart treatment encompassing the entire spectrum of LVEF [113]. The restoration of tubuloglomerular feedback is of pathophysiological importance. This causes vasoconstriction of the vas afferens and vasodilation of the vas efferens. Both together lead to a reduction in intraglomerular pressure and, thus, renal hyperfiltration. This results in the initial and reversible “dip” in the estimated glomerular filtration rate (eGFR) in patients subjected to SGLT2i. However, the renoprotective effects of SGLT2i have been clearly proven in several large trials addressing CKD, both in diabetic and non-diabetic patients [120,121,122]. SGLT2i may also be involved in the prevention and correction of anemia, possibly by erythropoiesis-stimulating effects [123]. A plethora of complex actions of SGLT2i (Figure 3) may be involved in these beneficial effects in chronic HF, spanning the continuum between HFrEF and HFpEF and ADHF in addition to its natriuretic and glucosuric effects, increase in ketogenesis contributing to beneficial metabolic effects on the failing cardiomyocytes, reversal of myocardial remodeling, reduction of albuminuria, and inhibition of inflammatory processes [94,124,125,126].

Another milestone of HF treatment in HFrEF patients with left bundle branch block (LBBB) is cardiac resynchronization therapy (CRT) [96]. In addition to its beneficial effects on LVEF, NYHA functional class, and heart failure-related hospitalizations, CRT has been shown to improve renal function by increasing cardiac output and increasing mean arterial pressure while decreasing central venous pressure [127]. CRT reduces SNS activity by decreasing adrenergic tone, which ultimately contributes to the improvement in renal function [128]. Mechanical circulatory support may be a treatment strategy to improve renal function and reduce diuretic resistance in CRS in advanced or terminal heart failure [129].

## 6. Conclusions

Complex network organ interactions of the heart and kidneys are intimately involved in the pathophysiology of cardiorenal syndrome (CRS), including interactions with the central nervous system (CNS). The five CRS types can be differentiated by pathophysiological pathways. While GFR remains the gold standard for assessing renal function, albuminuria is associated with an increased risk of mortality, heart failure-related hospitalizations, and clinical, echocardiographic, and circulating biomarkers of congestion. However, the precise clinical role of novel biomarkers, which may contribute to our pathophysiological understanding of AKI and cardiorenal disease, remains to be determined. Imaging modalities progress to comprehensive approaches encompassing the evaluation of kidneys, the heart, and the CNS. Functional magnetic resonance imaging of the kidneys may prove decisive for the detection of early changes in AKI or for the prediction of the progression of CKD. The prognostic impact of CRS among patients with COVID-19 infection highlights the need for special assessment and management of CRS in patients with certain comorbidities. Several mainstays of heart failure medication (ACEI, ARB, ARNI, MRA, SGKT2i) also have positive effects on the progressive course of CKD, again highlighting the necessity of differentiation between acute effects of glomerular pressure and true kidney injury or progression of kidney disease with tubulointerstitial fibrosis. As a novel therapeutic agent, the non-steroidal selective MRA finerenone has shown beneficial effects on cardiovascular and renal events in patients with diabetic nephropathy and albuminuria.

## Figures and Tables

**Figure 1 jcm-11-07041-f001:**
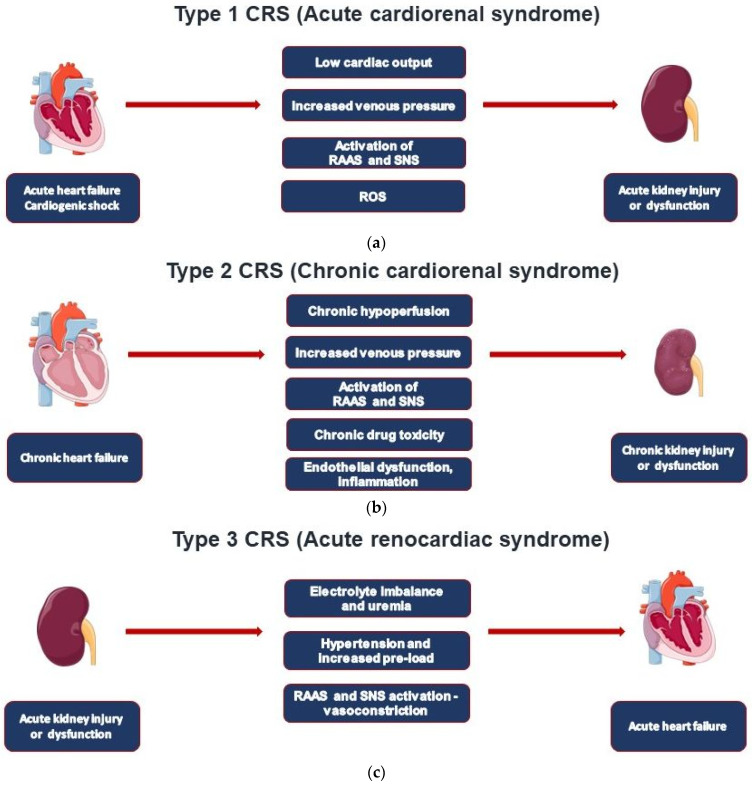
Pathogenic pathways involved in the five CRS types.

**Figure 2 jcm-11-07041-f002:**
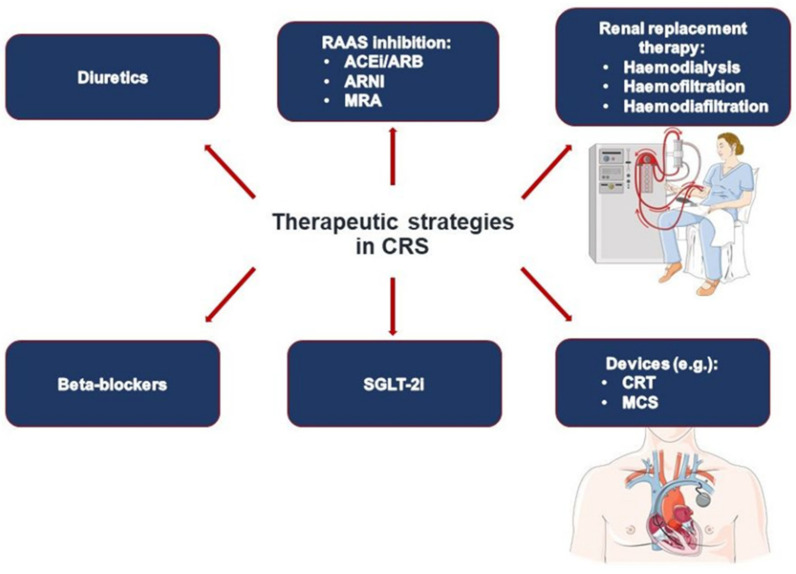
Therapeutic strategies in CRS.

**Figure 3 jcm-11-07041-f003:**
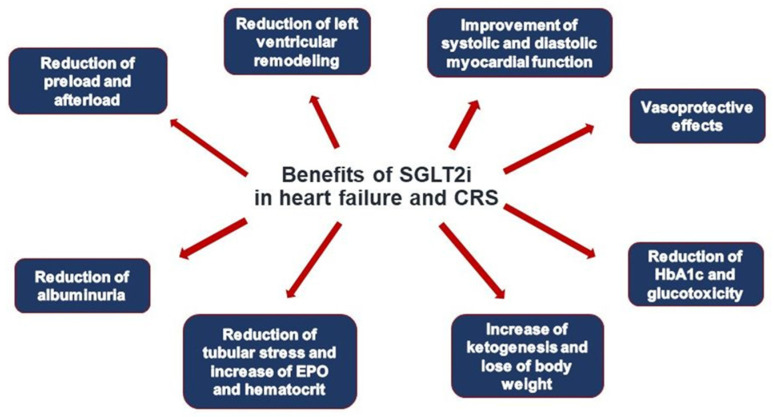
Benefits of SGLT2i in heart failure and CRS.

**Table 1 jcm-11-07041-t001:** Classification and basic characteristics of cardiorenal syndrome (CRS). AHF: acute heart failure; AKI: acute kidney injury; ACS: acute coronary syndrome; CHF: chronic heart failure; CKD: chronic kidney disease.

CRS Types	Mechanisms	Clinical Conditions
Type 1—Acute cardiorenal syndrome	AHF leading to AKI	AHF, ACS, cardiogenic shock
Type 2—Chronic cardiorenal syndrome	CHF leading to CKD	CHF regardless of cause
Type 3—Acute renocardiac syndrome	AKI leading to AHF	Volume overload, uremic metabolic disturbances, and inflammatory eruption
Type 4—Chronic renocardiac syndrome	CKD leading to CHF	CKD-induced cardiomyopathy resulting in cardiac remodeling and heart failure
Type 5—Secondary cardiorenal syndrome	Systemic disorder leading to cardiorenal dysfunction	Sepsis, diabetes, liver cirrhosis, amyloidosis, M. Fabry

## Data Availability

Not applicable.

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
