# Peer review of "Heart Failure and Cardiorenal Syndrome: A Narrative Review on Pathophysiology, Diagnostic and Therapeutic Regimens—From a Cardiologist’s View"

_jcm, 2022, doi:10.3390/jcm11237041_

Round 1

Reviewer 1 Report

Dear authors you will find below my detailed comments.   Q1 : The article is listed as a review but it¨s not mentioned in the title. Is it a critical review ? Something else?Please clarify or modify. Q2 : The article does not have a review structure and is quite extensive. You should better organize the structure,so you can have a “logical” presentation leading to results and conclusions. Q3 : It is a well written article with significant novelty and very useful information for the reader however you should made it more concise in my opinion. Q4 : The figures are really useful, please add the references that already exist in the text. Q5 : There are some typos erros , please correct them . eg line 153 covid etc General comment : The review is really extensive and its strucure needs to be improved. I would suggest you make it more concise and comprehensive.  I would recommend you to revise it and make it more concise and structured.

Author Response

Reviewer 1:

Dear authors you will find below my detailed comments.

Q1 : The article is listed as a review but it¨s not mentioned in the title. Is it a critical review ? Something else?Please clarify or modify.

Response: We thank the reviewer for this issue. We have now added in R1 that it is a narrative review. We also added “- From a cardiologist's view”, as suggested by the Editor.

Q2 : The article does not have a review structure and is quite extensive. You should better organize the structure,so you can have a “logical” presentation leading to results and conclusions.

Response:  We thank the reviewer for this suggestion. We have now reorganized the manuscript accordingly. A conclusion section has been added.

Q3 : It is a well written article with significant novelty and very useful information for the reader however you should made it more concise in my opinion.

Response: We appreciate this comment by the reviewer.

Q4 : The figures are really useful, please add the references that already exist in the text.

Response: We thank the reviewer for appreciating the usefulness of the figures. However, we cannot follow the suggestion to add references, since these figures are completely new. Adding references to the figures might implicate that we have adopted the figures from other publications, which is not the case.

Q5 : There are some typos erros , please correct them . eg line 153 covid etc

Response: We thank the reviewer for this important comment. We did correct some typos by proofreading the revised manuscript.

General comment : The review is really extensive and its strucure needs to be improved. I would suggest you make it more concise and comprehensive.  I would recommend you to revise it and make it more concise and structured.

Response: We thank the reviewer for this suggestion. We have now reorganized the manuscript accordingly.

Reviewer 2 Report

In this review, the authors cover cardiorenal syndrome, focusing on types 1 and 2, and provide some up-to-date information on different diagnostic and therapeutic modalities. The review provides a well-rounded overview of CRS; however, some more recent evidence and understanding of the pathophysiology and diagnostic testing is left out.

Comments:

-There is an omission in the discussion of pathophysiology of CRS type 1.  Most of the recent evidence shows a lack of kidney tubular injury in CRS, questioning the presence of true kidney injury at all, and that most creatinine changes are related to hemodynamic effects at the glomerulus. This is partially touched on with discussions of cardiac index and congestion, but not fully discussed to explain the often reversible and benign nature of these hemodynamic changes.

-In line with this comment, when discussing CRS type 2 starting on line 254, the authors should be explicit that many HF medications, ACE/ARB/ARNI and SGLT-2 predominately, acutely reduce GFR from reduced glomerular pressure, but this is a hemodynamic effect and over the long-term the decline in GFR either slows or remains unchanged from natural course. Again, an important distinction between hemodynamic effects at glomerulus and true kidney injury or progression of kidney disease with tubulointerstitial fibrosis. 

-The authors present data on the implications of cardiorenal syndrome in COVD-19 infection; however, this would be CRS type 5 as the SIRS response and potential direct toxicities of viral infection lead to injury of heart and kidney. The authors should be explicit in their description that these processes may not play a role in CRS types 1 and 2.

-in discussing albuminuria, citation of this recent article is timely DOI: 10.1093/eurheartj/ehac528

-for the statement on line 284 that Cystatin C “is a marker of proximal renal tubule injury”, this should be qualified that urine Cystatin C is a marker of proximal tubule injury/dysfunction since Cystatin C is fully metabolized in proximal tubule except with injury or dysfunction. Blood cystatin C only reflects glomerular filtration.

-This sentence is out of place in the middle of the discussion of Cystatin C: “Kidney 288 injury molecule-1 (KIM-1) is a protein that is detectable in the urine after ischemic or 289 nephrotoxic insult to proximal tubular cells and it appears to be highly specific for is- 290 chemic AKI (44)”

-the discussion of novel kidney biomarkers is outdated with numerous studies showing no association of tubular injury with outcomes with CRS in ADHF. DOI: 10.1016/j.jacc.2016.06.055, 10.1016/j.cardfail.2019.05.009, 10.1002/ejhf.1642, 10.1161/CIRCULATIONAHA.117.030112, 10.3390/ijms18071470, 10.1161/CIRCHEARTFAILURE.118.005552

-it is worth noting that ACE-I do not slow GFR decline in HFrEF (DOI: 10.1053/ j.ajkd.2019.05.010)

-the manuscript would benefit from a conclusion section with a paragraph summarizing highlights of the review.

-typo on line 154, the ‘D’ is left off of COVID-19

-typo on line 212, myocardial infarction is currently written as ‘myocardial infection’

Author Response

In this review, the authors cover cardiorenal syndrome, focusing on types 1 and 2, and provide some up-to-date information on different diagnostic and therapeutic modalities. The review provides a well-rounded overview of CRS; however, some more recent evidence and understanding of the pathophysiology and diagnostic testing is left out.

Response: We acknowledge this comment. On the other hand, the review needs to keep concise format, and cannot cover all issues of the literature, being a narrative and critical review. We would certainly add selected issues on recent evidence and understanding of the pathophysiology and diagnostic testing specified by the reviewer.

Comments:

-There is an omission in the discussion of pathophysiology of CRS type 1.  Most of the recent evidence shows a lack of kidney tubular injury in CRS, questioning the presence of true kidney injury at all, and that most creatinine changes are related to hemodynamic effects at the glomerulus. This is partially touched on with discussions of cardiac index and congestion, but not fully discussed to explain the often reversible and benign nature of these hemodynamic changes.

Response: We thank the reviewer for this important comment. We have now added these issues to pathophysiology of CRS type 1.

-In line with this comment, when discussing CRS type 2 starting on line 254, the authors should be explicit that many HF medications, ACE/ARB/ARNI and SGLT-2 predominately, acutely reduce GFR from reduced glomerular pressure, but this is a hemodynamic effect and over the long-term the decline in GFR either slows or remains unchanged from natural course. Again, an important distinction between hemodynamic effects at glomerulus and true kidney injury or progression of kidney disease with tubulointerstitial fibrosis. 

Response: We are indebted to the reviewer for this important issue, which has now been implemented in the revised manuscript.

-The authors present data on the implications of cardiorenal syndrome in COVD-19 infection; however, this would be CRS type 5 as the SIRS response and potential direct toxicities of viral infection lead to injury of heart and kidney. The authors should be explicit in their description that these processes may not play a role in CRS types 1 and 2.

Response: We thank the reviewer for this important suggestion. The section on pathophysiology encompasses also type 5 CRS, certainly. We have now added this clarification to the corresponding subsection of the section on pathophysiology.

-in discussing albuminuria, citation of this recent article is timely DOI: 10.1093/eurheartj/ehac528

Response: We cordially thank the reviewer for this important suggestion. We have now discussed this evidence in the revised manuscript.

-for the statement on line 284 that Cystatin C “is a marker of proximal renal tubule injury”, this should be qualified that urine Cystatin C is a marker of proximal tubule injury/dysfunction since Cystatin C is fully metabolized in proximal tubule except with injury or dysfunction. Blood cystatin C only reflects glomerular filtration.

Response: We have now adopted this important issue by reviewer 2.

-This sentence is out of place in the middle of the discussion of Cystatin C: “Kidney 288 injury molecule-1 (KIM-1) is a protein that is detectable in the urine after ischemic or 289 nephrotoxic insult to proximal tubular cells and it appears to be highly specific for is- 290 chemic AKI (44)”

Response: We thank the reviewer for this important suggestion. We have now reorganized this sentence to a more meaningful position in the text body.

-the discussion of novel kidney biomarkers is outdated with numerous studies showing no association of tubular injury with outcomes with CRS in ADHF. DOI:

10.1016/j.jacc.2016.06.055,

10.1016/j.cardfail.2019.05.009,

10.1002/ejhf.1642,

10.1161/CIRCULATIONAHA.117.030112,

10.3390/ijms18071470,

10.1161/CIRCHEARTFAILURE.118.005552

Response: We thank the reviewer for this highly important issue. We have added this novel evidence, and have shortened the section related to novel kidney biomarkers.

-it is worth noting that ACE-I do not slow GFR decline in HFrEF (DOI: 10.1053/ j.ajkd.2019.05.010)

Response: This important issue has now been added to the revised manuscript.

-the manuscript would benefit from a conclusion section with a paragraph summarizing highlights of the review.

Response: Thank you, we have now added a conclusions section the revised manuscript.

-typo on line 154, the ‘D’ is left off of COVID-19

Response: Thank you, we have solved this issue in the current version.

-typo on line 212, myocardial infarction is currently written as ‘myocardial infection’

Response: We thank the reviewer for this precise recommendation. This issue has now been corrected in the revised manuscript.

Reviewer 3 Report

The review is important and adequate to discuss CRS and CVS. The text is well written and all informations are important to this review. However, the text doesn't have a logical presentation. First, the authors present an introduction, pathophisiology, epidemiology, diagnosis, imaging of the heart and kidnet etc.

The text is not fluid. I suggest a re-organization of all topis. Moreover, the item subtitles are not adequate. A suggest a more accurate subtitle. Daingosis, pathophisiology is too general.

Second, the author did not mention about epigenetics factors, extracelullar vesicles etc It is important to bring the "stare of art"in this kind of review.

Finally, the figures are unttractive. The authors must improve the layout, information inside the figures. The table must include some referenfes also. 

Author Response

The review is important and adequate to discuss CRS and CVS. The text is well written and all informations are important to this review. However, the text doesn't have a logical presentation. First, the authors present an introduction, pathophisiology, epidemiology, diagnosis, imaging of the heart and kidnet etc.

Response: We thank the reviewer for this suggestion. We have now reorganized the manuscript accordingly.

The text is not fluid. I suggest a re-organization of all topis. Moreover, the item subtitles are not adequate. A suggest a more accurate subtitle. Daingosis, pathophisiology is too general.

Second, the author did not mention about epigenetics factors, extracelullar vesicles etc It is important to bring the "stare of art"in this kind of review.

Response: We thank the reviewer for this comment. We have now reorganized the manuscript accordingly.

Finally, the figures are unttractive. The authors must improve the layout, information inside the figures. The table must include some referenfes also. 

Response: We do not agree with this comment of reviewer 3. In line with reviewer 1 (Q4), we have the notion that the figures are useful, and therefore opt to keep these illustrative figures as they were initially submitted.

Round 2

Reviewer 3 Report

The manuscript was re-structured, organized and became more fluid. Tha authors adressed all questions and points.